# Memory, Emotion, and Quality of Life in Patients with Long COVID-19

**DOI:** 10.3390/brainsci13121670

**Published:** 2023-12-01

**Authors:** Katrina Espinar-Herranz, Alice Helena Delgado-Lima, Beatriz Sequeira Villatoro, Esther Marín Garaboa, Valeria Silva Gómez, Leonela González Vides, Jaime Bouhaben, María Luisa Delgado-Losada

**Affiliations:** 1Experimental Psychology, Cognitive Processes and Speech Therapy Department, Faculty of Psychology, Complutense University of Madrid, Campus de Somosaguas, 28223 Pozuelo de Alarcón, Spain; kespinar@ucm.es (K.E.-H.); alicedel@ucm.es (A.H.D.-L.); bsequeir@ucm.es (B.S.V.); esthmari@ucm.es (E.M.G.); valsilva@ucm.es (V.S.G.); jaimebou@ucm.es (J.B.); 2Optometry and Vision Department, Faculty of Optics and Optometry, Complutense University of Madrid, C. de Arcos de Jalón, 118, 28037 Madrid, Spain; leonelag@ucm.es; 3Group of Neurosciences, Psychoneuroendocrinology, Neuroimaging and Molecular Genetics in Neuropsychiatric Diseases, Instituto de Investigación Sanitaria San Carlos (IdISSC), Hospital Clínico de Madrid, 28040 Madrid, Spain

**Keywords:** chronic post-COVID syndrome, cognitive impairment, emotional impact, post-intensive care syndrome, quality of life

## Abstract

**(1) Background:** Persistent COVID is characterized by the presence of fatigue, mental fog, and sleep problems, among others. We aimed to study cognitive abilities (attention, executive functions, memory, language) and psychological and emotional factors in a group of participants of the population with persistent COVID-19 and asymptomatic or non-COVID-19-infected patients; **(2) Methods:** A total of 86 participants aged 18 to 66 years (X = 46.76) took part in the study, with 57 individuals (66.27%) in the experimental group and 29 (33.73%) in the control group. A comprehensive assessment included neuropsychological evaluations, evaluations of anxious and depressive symptomatology, assessments of the impact of fatigue, sleep quality, memory failures in daily life, and the perceived general health status of the participants; **(3) Results:** significant differences between groups were found in incidental learning within the Key Numbers task (U = 462.5; *p* = 0.001; *p* = 0.022) and in the Direct Digit Span (U = 562; *p* = 0.022), but not in the Inverse Digit Span (U = 632.5; *p* = 0.105). Differences were also observed in the prospective memory task of the Rivermead Prospective Memory Tasks (from the Rivermead Behavioural Memory Test) in the recall of quotations (U = 610; *p* = 0.020) as well as in the recall of objects (U = 681.5; *p* = 0.032). Concerning the task of verbal fluency, significant differences were found for both phonological cues (*p*- and s-) (t = −2.190; *p* = 0.031) and semantic cues (animals) (t = −2.277; *p* = 0.025). In terms of the psychological impact assessment, significant differences were found in the emotional impact across all variables studied (fatigue, quality of sleep, memory lapses, and the perceived general health status), except for quality of life; **(4) Conclusions:** Our results suggest that the sequelae derived from persistent COVID may have an impact on people’s lives, with higher levels of anxiety and depression, worse sleep quality, a greater number of subjective memory complaints, and a greater feeling of fatigue and impact on quality of life. Furthermore, poorer performance was observed in memory and verbal fluency.

## 1. Introduction

The global coronavirus disease 2019 (COVID-19) pandemic, caused by the novel coronavirus SARS-CoV-2, has not only presented acute health challenges but has also raised concerns about persistent symptoms in individuals long after the resolution of their initial infection. Known as “Long COVID” or post-acute sequelae of COVID-19 (PASC), this medical condition is characterized by a diverse range of symptoms that persist for more than 12 weeks after a confirmed or probable SARS-CoV-2 (COVID-19) infection [1,2] and affects not only physical health but also aspects of daily life [3]. These symptoms encompass neurological and cognitive impairment [3,4], fatigue [4,5,6,7], pain manifestations [6,7,8,9,10,11], cardio-pulmonary symptoms [1,11,12,13,14,15,16,17,18,19,20], anosmia–dysgeusia [1,17,19,20,21,22,23,24,25], and headache [1,13,18,20,22,23,24,25,26]. PASC is formally recognized as a consequence of COVID-19 and is colloquially referred to as “Long COVID” and “Post-COVID syndrome”. The World Health Organization (WHO) has established clinical case criteria for PASC, specifying that it typically occurs three months after the onset of COVID-19 symptoms, lasts for at least two months, and cannot be attributed to an alternative diagnosis [2].

Despite extensive research on the physical manifestations of Long COVID, limited attention has been given to the emotional, psychological, and cognitive dimensions of this condition. The study by Crivelli et al. [27] involved 45 post-COVID-19 participants and 45 controls who underwent neuropsychological evaluation. Significant differences were found between the groups in cognitive domains such as memory, attention, executive functions, and language. The study found that self-reported anxiety was associated with cognitive dysfunction in COVID-19 subjects. Although there are limited studies that evaluate specific cognitive domains, the preliminary findings suggest that executive function, memory, and attention are the domains that frequently exhibit disparities between patients with persistent COVID-19 symptoms and healthy controls up to 3 months after the illness [27,28,29,30,31].

Similarly, Ani Nalbandian et al. [28] state in their work that those suffering from PASC may manifest difficulties with concentration, memory, receptive language, and/or executive function. In addition to this, Beaud et al. [30] stated that in their study, composed of a sample of 13 participants, they found two cognitive profiles characteristic of the acute post-critical phase of severe COVID-19 that would remain present after hospital discharge in 70% of individuals: (1) normal MoCA score, but a tendency towards lower performances in executive functions than in other cognitive functions; (2) mild tosevere MoCA deficits with extensive cognitive impairment in executive, memory, attentional, and visuospatial functions, but relatively preserved orientation and language, with executive dysfunction. They studied mood and anxiety disturbances but not other variables that may affect cognitive profiles such as perceived quality of life, subjective perception of memory loss, sleep quality, or fatigue. The same applies to a study by Ortelli et al. [31] in a sample of 12 participants, whose results were that those with PASC have higher perceived fatigue and poorer performance on the MoCA test, meaning that their executive functions may be impaired.

This study addresses this gap by investigating the cognitive abilities—attention, executive functions, memory, and language—alongside psychological factors such as anxiety and depression in individuals with persistent COVID-19 symptoms. Additionally, sleep quality and fatigue will be explored, comparing those with ongoing symptoms to asymptomatic individuals or those who did not experience the acute phase of COVID-19. This research aims to provide a comprehensive understanding of the cognitive and psychological aspects, sleep patterns, and fatigue levels in individuals affected by long COVID-19, including those who may have been asymptomatic or never encountered the acute phase of the illness. Furthermore, this study describes the symptoms and their intensity, as well as their pattern based on participant measures. By delving into these dimensions, this study aspires to offer insights contributing to a more holistic approach to assessing and managing the enduring effects of COVID-19 on both symptomatic and asymptomatic populations.

## 2. Materials and Methods

### 2.1. Participants

The participant group comprised 57 individuals with a previous medical diagnosis of PASC and 29 control subjects who had either passed through the disease asymptomatically or had not contracted COVID-19. The primary inclusion criteria encompassed individuals between the ages of 18 and 66 years. In both cases, all participants were required to be free from the presence of other autoimmune diseases or chronic diseases and conditions, have no significant visual or auditory impairments that would hinder the completion of the tests, exhibit no language-related impairments that might impede comprehension or expression, and have no prior diagnoses of dementia or psychiatric disorders. Exclusion criteria encompassed minors, individuals lacking informed consent, and those failing to meet the aforementioned health criteria.

Participants included in the PASC group were recruited through referrals from the Association of patients with persistent COVID (Asociación de pacientes con COVID persistente) (https://covid-persistente.org/25 October 2023), while the participants included in the control group are from the Complutense University of Madrid.

### 2.2. Data Collection

#### 2.2.1. Neuropsychological Assessment

A comprehensive neuropsychological evaluation was conducted to assess cognitive functioning in all participants. Each assessment was chosen based on its established validity, reliability, and relevance to the specific domains under investigation. Another criterion was that these tests should have a standardized and validated version for the Spanish population. The selected tools included:-*Montreal Cognitive Assessment* (MoCA) [32]: Providing a global assessment of cognitive function.-*Wechsler Memory Scale III* (WMS III) Word List [33]: Focusing on verbal memory capabilities.-*Rey*–*Osterrieth Complex Figure Test* [34]: Evaluating visual–spatial and organizational skills.-*Wechsler Memory Scale III* (WMS III) *Digit Span Forward and Backward Test* [33]: Assessing attention and working memory.-*Trail Making Test A and B* [35]: Examining visual–motor processing speed and cognitive flexibility.-*Letter Cancellation Task, WAIS-III Digit Symbol Coding* [33]: Measuring attention, speed, and executive functions.-*Verbal Fluency Task, Boston Naming Test* [36]: Assessing language-related cognitive abilities.-*Rivermead Prospective Memory Tasks (from the Rivermead Behavioural Memory Test)* [37]: Exploring prospective memory abilities.

#### 2.2.2. Psychological Assessment

The psychological assessments were designed to capture a holistic view of participants’ psychological and emotional well-being. The selection of self-report questionnaires was informed by their psychometric properties and relevance to the study’s focus. Another criterion was that these tests should have a standardized and validated version for the Spanish population. The chosen assessments included:-*Modified Fatigue Impact Scale* (MFIS) [38]: Examining the impact of fatigue on daily functioning.-*Beck Depression Inventory* (BDI-2) [39]: Assessing the severity of depressive symptoms.-*State*–*Trait Anxiety Inventory* (STAI) [40]: Differentiating between state and trait anxiety.-*Short Form-12 Health Survey* (SF-12) [41]: Measuring overall health-related quality of life.-*Memory Failures of Everyday* (MFE) [42]: Exploring everyday memory lapses.-*Oviedo Sleep Questionnaire* [43]: Assessing sleep patterns and quality.

#### 2.2.3. Procedure

The study was conducted from June 2022 to May 2023 and received ethical approval from the Ethics Committee of the Clínico San Carlos University Hospital (Ref. 22/054-E). The research methods adhered to the ethical guidelines set forth by the research committee and the ethical principles outlined in the 1975 Declaration of Helsinki by the World Medical Association.

Access to participants’ medical records for research purposes followed established protocols. All participants were provided with and signed an informed consent form before participating in the study with all the information regarding the testing, and received individualized reports of their assessment results. Confidential measures were taken in order to respect the participants’ rights and privacy. Neuropsychological assessments took place in a quiet, well-lit environment of the Faculty of Medicine at the Complutense University of Madrid. Testing was completed in a single day, with an average duration of one hour per participant. Emotional state assessments were self-reported by participants via a Google Form questionnaire, with a follow-up to ensure completion. Individualized reports were provided to all participants, including recommendations for improving their well-being when necessary.

### 2.3. Analysis

The current study employed a non-experimental, cross-sectional design, as it did not involve the manipulation of independent variables or the random assignment of participants to experimental or control groups. Statistical analyses were conducted using the IBM SPSS Statistics^®^ version 25 (IBM Corp., Armonk, NY, USA) for MacOS Mojave 10.14.6. The data analysis process included the following steps:(1)*Assumption checks:* Prior to conducting the main analyses, the normality of the data distribution was assessed through the Shapiro–Wilk test for the control group (*n* = 29) and through the Kolmogorov–Smirnov test for the experimental group (*n* = 57) to ensure that the data met the assumptions necessary for parametric statistical tests (Appendix A).(2)*Descriptive analysis:* Statistical comparisons, using Student’s t-test for independent samples and the Mann–Whitney U test for the two independent samples, were conducted to quantify the variables related to cognitive abilities and psychological factors (anxiety, depression), quality of sleep, and fatigue of patients diagnosed with PASC versus asymptomatic patients and/or non-COVID-19-infected patients.

A significance level of *p* < 0.05 was established to determine whether the observed differences between groups were statistically significant. The results of this analysis provide insight into the potential impact of PASC on cognitive function, psychological well-being, fatigue, sleep quality, everyday memory failures, and perceived overall health.

## 3. Results

In the following sections, we present the results pertaining to demographic, cognitive, and psychological variables.

### 3.1. Results Pertaining to Demographic Variables

A total of 86 individuals aged between 18 and 66 participated in the study, with 57 participants in the experimental group and 29 participants in the control group. For a deeper description of the sample, please see Table 1.

It is worth noting that when comparing data between the experimental and control groups, statistically significant differences were observed only in the gender variable, indicating that the groups were not significantly different in terms of age, marital status, educational level, or occupational status.

### 3.2. Results Pertaining to Cognitive Variables

During the assessment period, there were instances of incomplete or missing data within specific cognitive tasks. In the experimental group *(n* = 55), two participants failed to correctly complete the prospective memory task regarding location and object. In the control group (*n* = 27), two individuals lacked time-related data in the copy, immediate recall, and delayed recall tasks of the Rey–Osterrieth Complex Figure Test. Additionally, in the control group (*n* = 28), one person did not complete the digit span task.

Regarding cognitive capacities, statistically significant differences were observed in the following variables:*Incidental Learning (Key Numbers Task):* Significant differences were found in incidental learning within the Key Numbers Task (U = 462.5, *p* = 0.001). This suggests variations in the ability to learn new information incidentally between the two groups.*Direct Digit Span:* Statistically significant differences were detected in the direct digit span (U = 562, *p* = 0.022), indicating variations in working memory capacity for forward digit span between groups.*Inverse Digit Span:* However, there were no statistically significant differences in the inverse digit span (U = 632.5, *p* = 0.105), suggesting that the backward working memory capacity was similar between the groups.*Rivermead Behavioral Memory Test (RBMT) Prospective Memory Tasks:* Significant differences were found in the RBMT prospective memory tasks for remembering appointments (U = 610, *p* = 0.020) and object recall (U = 681.5, *p* = 0.032). However, no significant differences were observed for location recall (U = 693.5, *p* = 0.082).*Verbal Fluency Task:* In both phonemic (*p*- and s-) (t = −2.190, *p* = 0.031) and semantic (animals) (t = −2.277, *p* = 0.025) verbal fluency tasks, statistically significant differences were noted, indicating variations in participants’ verbal fluency between the groups (see Figure 1).

It should be emphasized that no statistically significant differences were found in the remaining cognitive tests administered (see Figure 2). For additional details, please refer to Table 2.

These results highlight specific cognitive domains where individuals with PASC exhibited differences compared to control subjects, shedding light on the cognitive implications of the condition.

### 3.3. Results Pertaining to Psychological Variables

During the evaluation period, there were instances of incomplete or missing data in the assessment of psychological impact for some participants. Specifically, five individuals (n = 82) did not complete any of the questionnaires related to psychological impact. Among these, one participant belonged to the experimental group (n = 56), and four participants were from the control group (n = 25). Additionally, one participant from the control group (n = 24) did not complete the anxiety state-trait questionnaire.

Regarding the evaluation of psychological impact, fatigue, sleep quality, everyday memory lapses, and perceived overall health, statistically significant differences were observed in the variables under study. However, it’s worth noting that no statistically significant differences were found in the SF-12 quality of life questionnaire (t = 0.040, *p* = 0.968).

These results indicate that individuals with PASC experienced significant differences in various psychological aspects compared to control subjects. The impact on factors such as fatigue, sleep quality, memory issues, and perceived health is evident and demonstrates the psychological implications of the condition (see Figure 3). For detailed information, please refer to Table 3.

## 4. Discussion

Based on the obtained results of this study, whose goal was to provide insights for individuals contending with the long-term consequences of COVID-19, the hypothesis supported by the majority of studies is that patients with lingering effects of PASC would exhibit poorer performance in memory, attention, language, and executive functions compared to asymptomatic individuals and non-COVID-19-infected patients [8,10,44,45,46,47,48,49,50,51], is partially supported. In the present study, statistically significant differences were found in memory and language function, but not in attention or executive functions. These findings suggest that PASC may have specific cognitive implications, particularly in the domains of memory and language, which aligns with the research conducted by Daroische et al. and Premraj et al. [52,53].

These results also align with a study conducted by Ferrucci et al. [54], which assessed neuropsychological functioning in COVID-19 patients. Their findings revealed that, at the 5-month mark, 60% of the patients exhibited deficits in at least one cognitive function. Specifically, 41% showed deficits in processing speed, 20% in long-term verbal memory, and 18% in visuospatial memory. Importantly, one year later, 50% of the patients still displayed some form of cognitive deficit, although there was significant improvement in the results.

The discrepancies in cognitive outcomes between the current study and Ferrucci et al. research may be attributed to several factors. These could include differences in the assessment methods, the specific cognitive measures used, the timing of assessments, and the distinct profiles of COVID-19 patients in each study.

In the same way, Zhou et al. [55] conducted a study with a sample of 29 participants in which they reported that only one measure of attention was associated with COVID-19 infection. More recently, Søraas et al. [56] reported that patients affected by COVID-19 have a higher incidence of self-reported memory complaints following COVID-19 infection but did not perform any objective cognitive testing to see if this corresponded with such a subjective perception of the study participants.

The observed cognitive deficits in memory and language are indicative of the importance of ongoing monitoring and tailored interventions for patients with PASC. Identifying and addressing these cognitive challenges can significantly impact a patient’s quality of life and functional abilities.

Furthermore, our findings are in alignment with those presented by Graham et al. [8], who investigated a sample of 100 SARS-CoV-2-positive patients. They documented the frequency of neurological symptoms, conducted cognitive assessments, and gathered self-reported quality-of-life assessments from participants. Similar to our findings, their study did not reveal statistically significant differences in other cognitive domains such as executive functions or attention.

In contrast, the study conducted by Delgado-Alonso et al. [47] demonstrated cognitive deficits in these areas with a frequency three times higher than in the control group. It is important to recognize that the discrepancies between these studies may be attributed to several factors, including variations in the specific cognitive tests used, the timing of assessments, and the unique characteristics of the patient populations involved.

Regarding the psychological and emotional impact, patients with lingering effects of PASC exhibited a more pronounced psychological impact concerning anxiety, depression, sleep quality, and fatigue compared to asymptomatic individuals or those who did not contract COVID-19. These findings are consistent with results in the scientific literature, which reported the prevalence of symptoms such as anxiety, followed by depression, mood instability, irritability, sleep difficulties, and challenges in performing daily activities [4,10,29,30,48,49,50,51,52,53,54,57,58,59,60,61,62,63].

In a study conducted by Calabria et al. [60], fatigue was present in 82.3% of the study participants, and, through regression analysis, it was found that overall fatigue was largely predicted by depression, anxiety, apathy, and performance in cognitive tests of working memory and sustained attention. This indicates that the greater the affective symptoms and attentional/executive deficits, the higher the fatigue questionnaire scores.

All these psychological challenges may be motivated by the awareness of deficits, declining health, and personal and occupational repercussions reported by the participants. Additionally, the limited availability of socio-sanitary resources for treatment and intervention, coupled with the prevailing misinformation about the condition and social stigmatization associated with PASC symptoms, contribute to these psychological distresses. These factors can exacerbate excessive worry due to feelings of uncertainty and rumination, potentially leading to a perception of loss of control.

The results related to the perceived quality of life among patients with PASC and asymptomatic individuals or non-COVID-19-infected patients did not reveal statistically significant differences between the two groups. This finding differs from the results reported in various studies where individuals with PASC more frequently experience a negative impact on their perception of Health-Related Quality of Life (HRQoL) compared to those without these symptoms [50,52]. This negative impact can affect their ability to perform basic activities of daily living and maintain social relationships.

However, it is noteworthy that the study by Vagheggini et al. [63], which investigated symptoms of anxiety, depression, insomnia, and quality of life in a group of patients three months after recovering from the acute phase of COVID-19, found that a considerable proportion of patients exhibited symptoms of post-traumatic stress disorder, moderate depressive symptoms, and clinical insomnia. Nevertheless, no statistically significant evidence was found regarding changes in the perception of quality of life compared to before COVID-19. Despite this, there appeared to be some concern about the possibility of contracting COVID-19 again.

The discrepancy in findings concerning the perceived quality of life may be due to variations in the timing of assessments, the specific measures used to assess HRQoL, and the unique characteristics of the patient populations in each study. It is crucial to recognize that the experience of PASC is multifaceted, and individual perceptions of quality of life may be influenced by various factors, including the severity of symptoms, the presence of comorbidities, and personal resilience.

This study is not without limitations. One of them is, the sample size in the control group. Furthermore, these results must be interpreted with caution because the participants in the experimental group were not distributed across severity levels of the COVID-19 infection they suffered. Conducting further research in this direction with larger samples can help uncover potential group differences that may have been masked in this study due to the limited size of the control group. Additionally, future investigations should take into account participants’ educational levels and occupational backgrounds to analyze cognitive test results comparatively between individuals with PASC and the control group, which could help discern the influence of these variables on the outcomes of cognitive tests and the impact of COVID-19 infection.

Other potential avenues for future research may involve comparing results with a control group divided into those who have not been infected by SARS-CoV-2 and those who have been infected but do not exhibit subsequent symptoms. This comparison could shed light on whether this condition influences cognitive outcomes.

Understanding the complex interplay between physical and psychological symptoms, as well as their impact on quality of life, is essential to providing tailored support and interventions for PASC patients. In addition, these findings emphasize the importance of patient-centered care, focusing not only on physical recovery but also on addressing the psychological and emotional aspects of living with PASC.

In conclusion, the variations in the perception of quality of life among individuals with PASC warrant further investigation and emphasize the need for comprehensive care strategies that consider the unique challenges faced by these patients, both in terms of their physical health and their psychological well-being.

## 5. Conclusions

These findings underscore the importance of long-term follow-up for patients with PASC, as well as the need to establish an intervention plan that includes neurocognitive training for enhancing attention, memory, and executive functions. Additionally, psychological support should be provided to mitigate the risk of increased depressive and/or anxious symptoms among individuals who exhibit them during the evaluation. Such interventions are also essential for maintaining emotional well-being among those who do not present such symptoms.

The study highlights the multifaceted nature of PASC, emphasizing the need for comprehensive care that addresses both cognitive and emotional challenges. Tailored interventions, cognitive rehabilitation, and psychological support can significantly impact the quality of life and overall well-being of individuals recovering from COVID-19, particularly those who continue to experience symptoms associated with PASC.

Finally, further research, including larger sample sizes and a more extended follow-up period, is crucial for a deeper understanding of the cognitive and emotional consequences of PASC, which will contribute to the development of more effective and targeted interventions for these individuals. Additionally, long-term follow-up studies are warranted to track the trajectory of cognitive recovery and any potential lingering deficits.

Understanding the multifaceted nature of post-COVID conditions, including PASC, and their intersections with post-ICU effects like PICS is a critical area of future research. Investigating the complex interplay between cognitive, emotional, and physical symptoms is essential for developing effective interventions and improving the quality of life for individuals recovering from COVID-19

## Figures and Tables

**Figure 1 brainsci-13-01670-f001:**
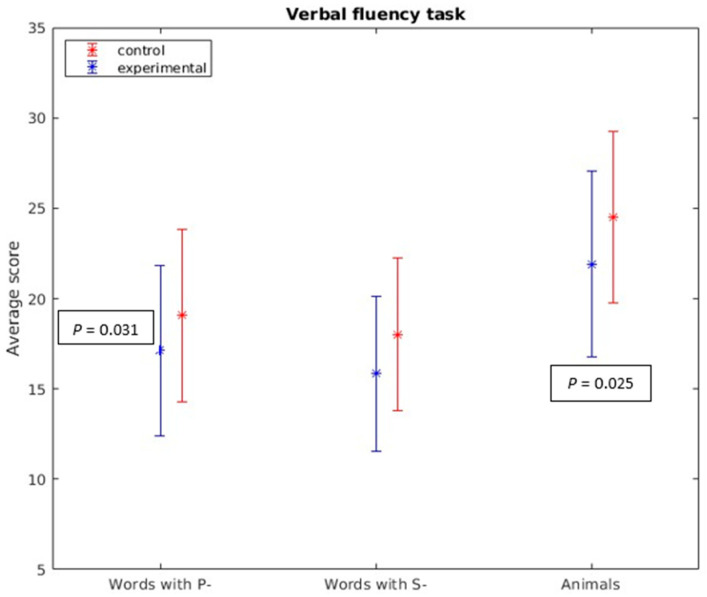
Verbal fluency task. In both phonemic (*p*- and s-) (t = −2.190, *p* = 0.031) and semantic (animals) (t = −2.277, *p* = 0.025) verbal fluency tasks, statistically significant differences were found.

**Figure 2 brainsci-13-01670-f002:**
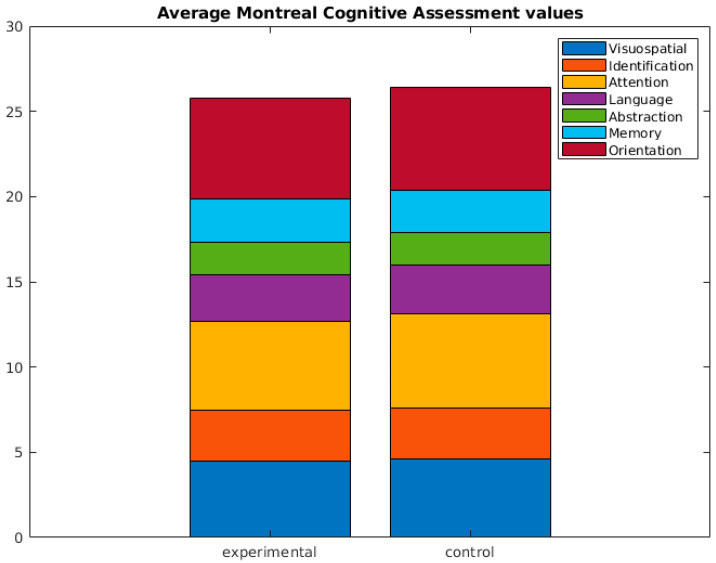
Average Montreal Cognitive Assessment (MoCA) values. No statistical differences were found.

**Figure 3 brainsci-13-01670-f003:**
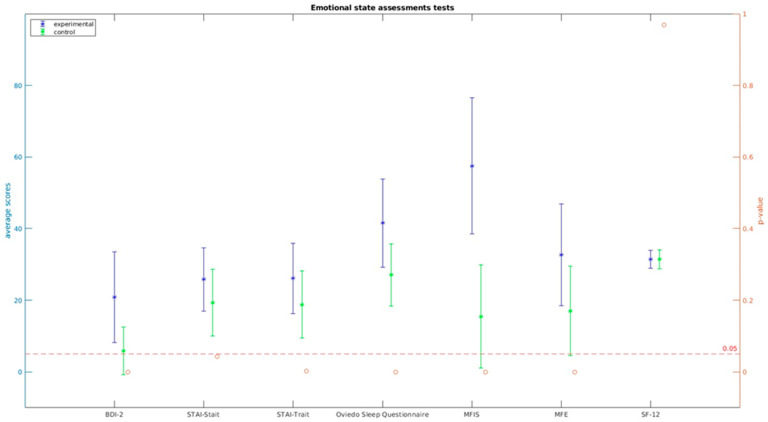
Emotional state assessment tests.

**Table 1 brainsci-13-01670-t001:** Sociodemographic characteristics of the participants.

*Sex*			
	Total n (%)	Experimental n (%)	Control n (%)
Man	38 (44.2)	17 (29.8)	21 (72.4)
Woman	48 (55.8)	40 (70.2)	8 (27.6)
		t-statistic: t = 4.605; *p* = 0.000 *	
*Age*	**Mean (SD)**	**Mean (SD)**	**Mean (SD)**
	46.76 (10.212)	48.06 (8.22)	44.22 (13.188)
		U-statistic: U = 664; *p* = 0.210	
*Marital status*			
	**Total** **n (%)**	**Experimental** **n (%)**	**Control** **n (%)**
Single	22 (25.6)	14 (24.6)	8 (27.6)
Married	43 (50)	28 (49.1)	15 (51.7)
Divorced	3 (3.5)	3 (5.3)	0 (0)
Cohabiting couple	11 (12.8)	7 (12.3)	4 (13.8)
No data	7 (8.1)	5 (8.8)	2 (6.9)
		U-statistic: U = 670; *p* = 0.714	
*Educational level*			
	**Total** **n (%)**	**Experimental** **n (%)**	**Control** **n (%)**
Primary school studies	4 (4.7)	1 (1.8)	n = 3 (10.3)
High school studies	20 (23.3)	15 (26.3)	n = 5 (17.2)
University studies	33 (38.4)	25 (43.9)	n = 8 (27.6)
Master’s/PhD	25 (29.1)	12 (21.1)	n = 13 (44.8)
No data	4 (4.7)	4 (7)	n = 0 (0)
*Employment*			
	**Total** **n (%)**	**Experimental** **n (%)**	**Control** **n (%)**
Non-qualified (homemakers’ collective)	1 (1.3)	0 (0)	1 (3.4)
Qualified for a manual trade (bricklayer, seamstress …)	7 (8.8)	3 (5.3)	4 (13.8)
Qualified for a non-manual trade (administrative, technician …)	24 (29.1)	21 (36.8)	4 (13.8)
Professionals (university workers)	38 (44.2)	23 (40.4)	15 (51.7)
Manager	5 (5.8)	4 (7)	1 (3.4)
Student	3 (3.5)	0 (0)	3 (10.3)
Retired	1 (1.2)	0 (0)	1 (3.4)
No data	6 (7)	6 (10.5)	0 (0)
		U-statistic: U = 633.5; *p* = 0.253	

* *p* < 0.001; t = Student’s *t*-test; U = Mann–Whitney U test.

**Table 2 brainsci-13-01670-t002:** Neuropsychological assessment tests average scores.

*Montreal Cognitive Assessment (MOCA)*				
		Experimental n = 57 Mean (SD)	Control n = 29 Mean (SD)	Statistic (U, p)
Visuospatial		4.49 (0.658)	4.59 (0.733)	732.5 (0.318)
Identification		2.98 (0.132)	3.00 (0.00)	812 (0.476)
Attention		5.19 (1.076)	5.52 (0.829)	672 (0.115)
Language		2.74 (0.518)	2.90 (0.409)	699.5 (0.078)
Abstraction		1.91 (0.285)	1.90 (0.310)	813.5 (0.813)
Memory		2.53 (1.324)	2.48 (1.595)	811 (0.885)
Orientation		5.91 (0.285)	6.00 (0.000)	754 (0.102)
Total		25.81 (2.057)	26.38 (2.321)	680 (0.174)
*List of words*				
		**Experimental** **(n = 57) ** **Mean (SD)**	**Control** **(n = 29) ** **Mean (SD)**	**Statistic** **(t, p)**
Immediate memory		31.58 (6.636)	31.07 (4.480)	0.372 (0.711)
Delayed memory		7.65 (3.044)	7.48 (2.721)	0.248 (0.805)
Recognition		22.56 (2.163)	22.90 (1.698)	750 (0.457)
*Rey* *–* *Osterrieth Complex Figure Test*				
		**Experimental** **(n = 57) ** **Mean (SD)**	**Control** **(n = 29) ** **Mean (SD)**	**Statistic** **(U, p)**
Copy	Time	132.29 (39.707)	n = 27; 129 (51.909)	0.311 (0.757)
	Score	33.68 (2.329)	33.52 (2.879)	806.5 (0.853)
Immediate memory	Time	108.46 (53.733)	n = 27; 95.44 (36.297)	711.5 (0.578)
	Score	21.09 (6.602)	21.26 (7.199)	807 (0.859)
Delayed memory	Time	82.44 (35.263)	n = 27; 71.07 (31.006)	634 (0.194)
	Score	19.96 (6.744)	21.38 (7.009)	719.5 (0.328)
*Digital Span Task*				
		**Experimental** **(n = 57) ** **Mean (SD)**	**Control** **(n = 29) ** **Mean (SD)**	**Statistic** **(U, p)**
Digit Span Forward Test		3.98 (1.077)	4.75 (1.481)	562 (0.022 *)
Digit Span Backward Test		2.84 (0.941)	3.18 (1.020)	632.5 (0.105)
Total Forward Digits		6.72 (2.374)	7.25 (2.675)	707.5 (0.393)
Total Backward Digits		4.75 (1.864)	5.07 (1.698)	690 (0.304)
*Trail Making Test (TMT)*				
		**Experimental** **(n = 57) ** **Mean (SD)**	**Control** **(n = 29) ** **Mean (SD)**	**Statistic** **(U, p)**
TMTa	Trials	23.95 (0.225)	24 (0)	783 (0.211)
	Errors	0.09 (0.285)	0 (0)	754 (0.102)
	Time	38.89 (14.495)	35.97 (14.618)	687 (0.202)
TMTb	Trials	22.74 (1.110)	22.17 (2.726)	769 (0.297)
	Errors	0.39 (1.176)	0.83 (2.578)	773 (0.447)
	Time	77.26 (30.181)	77.86 (49.713)	689 (0.209)
*Cancellation Task*				
		**Experimental** **(n = 57) ** **Mean (SD)**	**Control** **(n = 29) ** **Mean (SD)**	**Statistic** **(U, p)**
TOT Effectiveness		285.95 (96.666)	290.41 (98.925)	755.5 (0.517)
CON Concentration index		18.25 (5.190)	20.48 (4.556)	t = −1.966 (0.053)
TR Total of responses		296.89 (92.640)	317.48 (74.601)	686.5 (0.201)
TA Total of trials		18.54 (5.352)	20.48 (4.672)	667 (0.144)
Commissions		0.11 (0.409)	0.10 (0.310)	802 (0.637)
Omissions		3.54 (3.576)	3.24 (2.116)	776 (0.641)
*Rivermead Behavioural Test (RMBT)*				
		**Experimental** **(n = 57) ** **Mean (SD)**	**Control** **(n = 29) ** **Mean (SD)**	**Statistic** **(U. p)**
Recalling the date		1.51 (0.691)	1.90 (0.724)	610 (0.020 *)
Recalling of object		1.82 (0.475)	2.00; (0)	693.5 (0.032 *)
Recalling of place		1.80 (0.487)	1.97; (0.186)	693.5 (0.082)
*Digit Symbol Coding*				
		**Experimental** **(n = 57) ** **Mean (SD)**	**Control** **(n = 29) ** **Mean (SD)**	**Statistic** **(U. p)**
Score		69.157 (17.034)	75.241 (14.032)	631 (0.074)
Incidental memory		4.67 (2.423)	6.59 (2.529)	462.5 (0.001 *)
*Boston Vocabulary Test*				
		**Experimental** **(n = 57) ** **Mean (SD)**	**Control** **(n = 29) ** **Mean (SD)**	**Statistic** **(U. p)**
Spontaneous answers		55.42 (3.635)	55.55 (2.910)	803.5 (0.832)
Semantic key		0.32 (0.540)	0.28 (0.528)	795 (0.709)
Phonological key		3.12; 2.673	2.38; 1.821	729; 0.367
*Verbal Fluency Task*				
		**Experimental** **(n = 57) ** **Mean (SD)**	**Control** **(n = 29) ** **Mean (SD)**	**Statistic** **(t. p)**
Words with *p*- (Spanish language)		17.11 (4.742)	19.07 (4.765)	−2.190 (0.031 *)
Words with *s*- (Spanish language)		15.84 (4.279)	18.00 (4.226)	
Animals		21.91 (5.149)	24.52 (4.741)	−2.277 (0.025 *)

* *p* < 0.05 t = Student’s *t*-test; U = Mann–Whitney U test.

**Table 3 brainsci-13-01670-t003:** Average scores of emotional state assessment tests.

	Experimental (n = 55) Mean (SD)	Control (n = 25) Mean (SD)	Statistic (U. p)
BDI-2	20.88 (12.684)	5.83 (6.627)	167.5 (0.0 **)
STAI-Stait	25.79 (8.800)	n = 24; 19.33 (9.375)	479.5 (0.043 *)
STAI-Trait	26.09 (9.823)	n = 24; 18.79 (9.413)	t = 3.082 (0.003)
Oviedo Sleep Questionnaire	41.55 (12.357)	27.08 (8.717)	226 (0.0 **)
MFIS	57.48 (19.012)	15.48 (14.353)	72 (0.0 **)
MFE	32.64 (14.189)	17.00 (12.437)	306.5 (0.0 **)
SF-12	31.46 (2.515)	31.44 (2.599)	0.040 (0.968)

* *p* < 0.05 ** *p* < 0.001 t = t-Student; U = U-Mann Whitney. BDI-2 Beck Depression Inventory; MFIS Modified Fatigue Impact Scale; MFE Memory Failures of Everyday; SF-12 Health Survey.

## Data Availability

Upon request to the corresponding author. The data are not publicly available due to containing information that could compromise the privacy of research participants.

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
