# Peer review of "Memory, Emotion, and Quality of Life in Patients with Long COVID-19"

_brainsci, 2023, doi:10.3390/brainsci13121670_

Round 1

Reviewer 1 Report

Comments and Suggestions for Authors

Dear Authors

I’ve provided the following revisions and recommendations. 

a. Abstract

The abstract could benefit from clearer and more concise language. The use of abbreviations such as "PASC" and "COVID-19" could be explained for readers who may not be familiar with the terminology. Additionally, the abstract could benefit from providing more specific information about the study's findings, such as the magnitude of the differences observed in cognitive abilities and psychological factors between the experimental and control groups.

b. introduction 

The introduction is overly verbose and lacks conciseness. It could benefit from a more direct and clear presentation of the main points. Additionally, the introduction does not provide a clear and strong thesis statement or research question, which is essential for guiding the reader and setting the tone for the rest of the study. hence, you can use dome sources regarding the issuse of Covid-19 and life style as the following references:

"Mental health, eating habits and physical activity levels of elite Iranian athletes during the COVID-19 pandemic." Science & Sports (2023).

The repetitive use of citations and references within the introduction also disrupts the flow of the text and could be better integrated into the main body of the paper. Overall, the introduction could benefit from a more focused and organized structure.

c. Methods

There are a few aspects of the "Materials and Methods" section that could be criticized: 

-It would be beneficial to include more detailed demographic information such as age, gender distribution, and any relevant medical history beyond the primary inclusion and exclusion criteria.

-The use of referrals from a specific association for patients with persistent COVID-19 may introduce selection bias into the participant group. This method of recruitment could potentially lead to a non-representative sample, as individuals who are members of this association may have different experiences and outcomes compared to the broader population of individuals with persistent COVID-19 symptoms.

-The rationale for selecting specific neuropsychological and psychological assessment tools is not provided. It would be beneficial to include a brief explanation of why these particular assessments were chosen and how they align with the study's research questions and objectives. This would enhance the transparency and rigor of the study methodology.

-Limited Information on Data Analysis: While the general approach to data analysis is outlined, there is a lack of detail regarding specific statistical methods and tests used to analyze the data. Providing more information on the statistical techniques employed, including any assumptions made and steps taken to address potential confounders, would enhance the methodological transparency and allow for a better understanding of the study's analytical approach.

-Ethical Considerations: While the study mentions receiving ethical approval and adhering to ethical guidelines, it would be beneficial to provide more specific details regarding participant consent procedures, confidentiality measures, and any potential conflicts of interest. This additional information would help to ensure that the study was conducted in an ethically sound manner.

d. Results

The figures can help the results better highlighted.

e. Discussion: focus more on study limitation

Author Response

We wanted to express our gratitude for taking time to review our manuscript entitled "Memory, Emotion, and Quality of Life in Patients with Long COVID-19". We strongly appreciate your interest in the research and your criticism and suggestions. Please find below a point-by-point responses for both review reports. We hope you now consider our manuscript suitable for publication.

Reviewer 2 Report

Comments and Suggestions for Authors

Thanks to the authors for sharing their manuscript. Appreciating the manuscript highly, I have a few small comments:

1.      I would like to see a more meaningful introduction. The authors describe the negative impact of the COVID-19 pandemic in two small paragraphs, but do not focus on the subject of the research. Moreover, a small introduction looks disproportionate together with a detailed discussion and conclusion.

2.      As far as I understand, the study was conducted in Spain. Have all the methods used in the study been adapted into Spanish? Please indicate this in the manuscript.

The experimental and control samples differ in size. I understand that this is due to the availability of respondents, but still I would like to see a preliminary calculation of the sample size. Was there enough sample for reliable results?

Author Response

(The authors gave the same response as above.)
